



# Estimation of particulate organic nitrates from thermodenuder-aerosol mass spectrometer measurements in North China Plain

Weiqi Xu[1], Masayuki Takeuchi[2], Chun Chen[1,3], Yanmei Qiu[1,3], Conghui Xie[1,3,a], Wanyun Xu[4], Nan Ma[5], Douglas R. Worsnop[6], Nga Lee Ng[2,7,8,*] and Yele Sun[1,3,9,*]

[1]State Key Laboratory of Atmospheric Boundary Layer Physics and Atmospheric Chemistry, Institute of Atmospheric Physics, Chinese Academy of Sciences, Beijing 100029, China
[2]School of Civil and Environmental Engineering, Georgia Institute of Technology, Atlanta, GA 30332, USA
[3]College of Earth and Planetary Sciences, University of Chinese Academy of Sciences, Beijing 100049, China
[4]State Key Laboratory of Severe Weather & Key Laboratory for Atmospheric Chemistry, Institute of Atmospheric
Composition, Chinese Academy of Meteorological Sciences, Beijing, 100081, China
[5]Institute for Environmental and Climate Research, Jinan University, Guangzhou 511443, China
[6]Aerodyne Research Inc., Billerica, Massachusetts 01821, USA
[7]School of Earth and Atmospheric Sciences, Georgia Institute of Technology, Atlanta, GA 30332, USA
[8]School of Chemical and Biomolecular Engineering, Georgia Institute of Technology, Atlanta, GA 30332, USA
[9]Center for Excellence in Regional Atmospheric Environment, Institute of Urban Environment, Chinese Academy of Sciences, Xiamen 361021, China
[a]now at: State Key Joint Laboratory of Environmental Simulation and Pollution Control, College of Environmental Sciences and Engineering, Peking University, Beijing 100871, China

*Correspondence*: Yele Sun (sunyele@mail.iap.ac.cn) and Nga Lee Ng (ng@chbe.gatech.edu)

**Abstract.** Particulate organic nitrates (pON) are an important component of secondary organic aerosol in biogenic emission dominant environments, and play a critical role in $NO_x$ cycles. However, estimation of pON has been a challenge in polluted environments, e.g., North China Plain with high concentrations of inorganic nitrate and $NO_x$. Here we developed a method for estimation of pON from the measurements of high-resolution aerosol mass spectrometer coupled with a thermodenuder based on the volatility differences between inorganic nitrate and pON. The results generally correlated well with those estimated 25 from positive matrix factorization of combined organic and inorganic mass spectra and from the ratio of $NO^+$ to $NO_2^+$ ($NO_x^+$ ratio), yet had improvements in reducing negative values due to the influences of high concentration of inorganic nitrate and constant $NO_x^+$ ratio of organic nitrates ($R_{ON}$). By applying this approach to the measurements at an urban (Beijing) and a rural site (Gucheng) in summer and winter in North China Plain, we estimated that the average mass concentrations of $NO_{3,org}$ (1.8 µg m⁻³ vs.1.0 µg m⁻³) and pON to OA (27.5% vs. 14.8%) were higher in summer than in winter in Beijing, indicating more 30 pON formation in biogenically and anthropogenically mixed environments. In addition, the average $NO_{3,org}$ loading in Gucheng was 1.9 µg m⁻³, and the pON at the rural site also showed higher contribution to OA than that in Beijing during wintertime due to higher primary emissions and gaseous precursors in Gucheng. In addition, $R_{ON}$ was determined and showed considerable differences between day-night and clean-polluted periods, highlighting the complexity of pON compounds from different chemical pathways (e.g., OH and $NO_3$ oxidation) and sources.



## Introduction

Particulate organic nitrates (pON), accounting for 5-77% of organic aerosol (OA) (Ng et al., 2017), are important atmospheric constituents as they affect $NO_x$ recycling, ozone, and secondary organic aerosol (SOA) production (Perring et al., 2013;Present et al., 2020). Most previous studies on pON quantification are based on filter measurements using gas chromatography (GC) (O'Brien et al., 1995) and Fourier transformed infrared (FTIR) spectroscopy (Blando et al., 1998). On-line measurement techniques, e.g., thermal dissociation laser induced fluorescence (TD-LIF) (Day et al., 2002), cavity ring-down spectroscopy (TD-CRDS) (Thieser et al., 2016) and cavity attenuated phase shift spectroscopy (TD-CAPS) (Sadanaga et al., 2016) were also used to characterize pON in the atmosphere. The principle is to convert pON to $NO_2$ first and then to subtract ambient $NO_2$ from the total $NO_2$. However, the pON may not be accurately quantified under ambient $NO_2$ dominated conditions. High-resolution time-of-flight chemical ionization mass spectrometer (HR-ToF-CIMS) using iodide-adduct ionization (Lee et al., 2014) with a filter inlet for gases and aerosols (FIGAERO) (Lopez-Hilfiker et al., 2014) (FIGAERO-CIMS) allows for simultaneous characterization of molecular compositions of gas and particle phases of ON. However, the lack of authentic ON standards and different instrument sensitivity for various ON compounds increase the challenges in quantification of pON.

Recently, pON were also quantified using high-resolution aerosol mass spectrometer (HR-AMS) mainly based on the differences of fragmentation patterns between pON and inorganic nitrate (Xu et al., 2015b;Farmer et al., 2010;Kortelainen et al., 2017;Ng et al., 2017). "PMF method" and "$NO_x$ method" are two most commonly used method for pON quantification (Fry et al., 2013;Xu et al., 2015b;Yu et al., 2019;Dai et al., 2019). The "PMF method" by combining organic and inorganic fragments in positive matrix factorization (PMF) analysis was able to identify organic aerosol factors that showed largely different $NO^+/NO_2^+$ ratio from pure ammonium nitrate (AN). The $NO_x$ fragments are then assumed to be mainly from pON (Xu et al., 2015b;Zhang et al., 2016). However, the separation of inorganic and organic $NO_x^+$ may be limited by the bilinear model depending on the correlations between inorganic nitrate and pON. In addition, some primary organic aerosol (POA) factors from "PMF method" also contained much higher $NO^+$ or $NO_2^+$ compared to directly measured mass spectrum of primary emissions (Xu et al., 2020) (Fig. S1), increasing the uncertainties in quantification of pON. The "$NO_x$ method" is based on the upper and lower bounds of $R_{ON}$ (ratio of $NO^+$ to $NO_2^+$ of pON) that were determined from oxidations of isoprene and monoterpenes (β-pinene) with nitrate ($NO_3$) radicals (Fry et al., 2009;Bruns et al., 2010;Boyd et al., 2015). One of the major uncertainties is that $R_{ON}$ varies largely depending on reaction conditions and precursors in ambient rather than being constant values. In some environments, the measured ratio of $NO^+$ to $NO_2^+$ in ambient is even lower than the average ratio of $NO^+$ to $NO_2^+$ from ammonium nitrate ($R_{AN}$), resulting in negative values of pON estimated by the "$NO_x$ method", especially for the wintertime in China when aerosol particles are dominated by inorganic nitrate (Yu et al., 2019;Zhu et al., 2016).

Despite the uncertainties above, extensive studies have been conducted to characterize spatial and seasonal variations of pON





in low NO$_x$ environments with high biogenic emissions, e.g., the southeastern United States and Europe (Kiendler-Scharr et al., 2016;Lee et al., 2016;Xu et al., 2015b;Ng et al., 2017). Similar work, however, is limited in China, especially in northern China with high emissions of NO$_x$ and anthropogenic volatile organic compounds (VOCs). In recent years, some studies explored the concentrations of pON at two background sites in spring (Zhu et al., 2016) and during biomass burning and coal

combustion periods in Beijing (Zhang et al., 2016). Yu et al. (2019) found a good correlation between pON and fresh SOA at night at an urban site in southern China based on "NO$_x$ method" and "PMF method". In contrast, Qiao et al. (2020) reported that pON were more correlated with the emission from biomass burning but poorly correlated with less-oxidized oxygenated OA (LO-OOA) at a rural site (Xianghe) in summer. Despite these results, our understanding of the levels, sources and processes of pON in North China Plain (NCP) remains poor.

In this study, we demonstrate the applications of thermodenuder (TD)-HR-AMS in characterization of pON (referred to as "TD-AMS method") during wintertime and summertime in NCP (including an urban site and a rural site) and compare pON mass loadings from three different methods, i.e., "NO$_x$ method", "PMF method" and "TD-AMS method". The mass loadings and diurnal variations of pON during one summer and two winter campaigns are elucidated and the impacts of relative humidity and temperature on pON are discussed.

**2 Experimental methods**

**2.1 Experimental set-up**

This study was performed at the Institute of Atmospheric Physics, an urban site in Beijing, and Gucheng, a rural site in Hebei province. The field observations at the urban site were conducted in summer from 20 May to 23 June and winter from 20 November to 25 December in 2018, and at the rural site from 10 December 2019 to 13 January 2020. The set up and operations

of TD and HR-AMS were similar to previous studies (Xu et al., 2019). The HR-AMS that was operated in V mode with a time resolution of 3 min was placed downstream of the TD. TD alternated between the bypass line and TD line in a cycle of 30 min. The TD heating temperature was set as 50, 70, 90, 109, 120, 127 and 250 ℃ in summer and 50, 70, 90, 109, 127, 150 and 250 ℃ in winter in Beijing, while in winter of 2019 in Gucheng, the setting TD temperature ramped linearly from 50 ℃ to 250 ℃. The average particle loss in the TD was corrected according to the calibration by using NaCl and was found to be 10%

(Huffman et al., 2008).

**2.2 Chemical data analysis**

The HR-AMS data analysis was performed using PIKA (V 1.62F). The HR-AMS was calibrated for ionization efficiency (IE) using pure ammonium nitrate by following the standard protocols (Jayne et al., 2000). Here, we apply three independent





methods of quantifying pON. The first method is "$NO_x$ method" which is based on the differences in $NO^+/NO_2^+$ ratio between organic and inorganic nitrates (Farmer et al., 2010;Xu et al., 2015b). The $R_{AN}$ was determined from ionization efficiency (IE) calibrations as 3.7 in summer and 3.4 in winter in both Beijing and Gucheng. However, we found that the winter $R_{AN}$ was higher than those from ambient observations. Therefore, we chose a period with high $NO_3$ loadings (assuming predominant

inorganic nitrate ($NO_{3,inorg}$) contribution) to determine $R_{AN}$ which was 2.8 and 2.2 , respectively in Beijing and Gucheng in winter. To reduce the effect of variation in $R_{ON}$ value among different instruments, we determined $R_{ON}$ by multiplying $R_{ON}/R_{AN}$ with $R_{AN}$ in using "$NO_x$ method", assuming that $R_{ON}/R_{AN}$ is independent of instrument (Fry et al., 2013). The $R_{ON}/R_{AN}$ ratios of 2.08 and 4.17 from isoprene and β-pinene oxidation experiments were used as upper and lower bounds of pON in this study (Xu et al., 2015b). Hence, $R_{ON}$ of 7 and 15, 5 and 12, 4 and 10 obtained in summer and winter in Beijing and Gucheng during

wintertime, respectively, were used to estimate the upper and lower bounds of pON loadings in this study.

The second method is based on PMF analysis (referred to "PMF method") (Yu et al., 2019;Xu et al., 2015b). The time series and mass spectra of PMF factors resolved from $PMF_{Org}$ and $PMF_{Org+NO3}$ were shown in detail in Figs. S2-S3. Note that the default relative ionization efficiencies (RIE) of 1.1 and 1.4 were applied for inorganic nitrate and $NO_{3,org}$, respectively, in this study. Considering the high mass fraction of ammonium nitrate, the composition-dependent collection efficiency (CDCE) was

applied to the bypass data (Middlebrook et al., 2012). In contrast, a constant CE of 0.5 was used for those from TD, consistent with the data processing in previous studies (Xu et al., 2016;Huffman et al., 2009). The elemental composition of OA was calculated using "Improved-Ambient (I-A)" method (Canagaratna et al., 2015). The OA factors representing different sources were identified by PMF and the detailed PMF analysis during three campaigns have been given in Xu et al. (in preparation) and Chen et al. (in preparation).

The third method denotes the "TD-AMS method" which is based on the different volatility between organic and inorganic nitrates. Briefly, $NO^+$ and $NO_2^+$ (hereafter $NO_x^+$) signals in HR-AMS mass spectra can be from fragmentation of both particulate organic and inorganic nitrates, whereas $C_xH_yN_z^+$ and $C_xH_yO_zN_p^+$ are completely from particulate organic nitrogen compounds including ON and other nitrogen-containing organic compounds. The $C_xH_yN_z^+$ and $C_xH_yO_zN_p^+$ dominated the N mass for non-ON organic nitrogen compounds (Fig. S4), while $NO^+$ and $NO_2^+$ were the main N-containing ions for ON (Farmer

et al., 2010;Boyd et al., 2015;Xu et al., 2015a). As shown in Fig.1, the N mass from $NO^+$ and $NO_2^+$ dominated the total N in ambient (>95%), indicating that pON dominated the total particulate organic nitrogen compounds in NCP in both summer and winter. Owing to higher volatility of $NO_{3,inorg}$ than ON (Huffman et al., 2008;Ng et al., 2017), the remaining mass loadings of nitrate fragments from pON (denoted as $NO_x^+{}_{,Org}$) is expected to be much higher than that of nitrate fragments from inorganic nitrates (denoted as $NO_x^+{}_{,inorg}$) under high TD temperatures (Huffman et al., 2008;Berkemeier et al., 2020). As shown in Fig.

1, $NO^+$ and $NO_2^+$ decreased significantly from 25 °C to 90 °C, whereas the decreases slowed down substantially for $T > 90$ °C.





Hence, we assumed that the $NO_{3,inorg}$ evaporated completely at $T = 90$ °C and the $NO_x^+$ measured by HR-AMS for $T > 90$ °C was the remaining organic nitrates compounds upon heating, consistent with previous studies reporting a complete evaporation of pure AN at ~60 °C (Huffman et al., 2008). The slightly different evaporation temperature of AN between this study and Huffman et al. (2008) could be due to the different aerosol mixing states in different environments.

5   Assuming that the volatility of pON can be represented by total organic nitrogen compounds, the mass concentrations of pON in ambient can be determined as:

$$[NO_{3,org}] = \frac{[NO_3]_{T=90°C}}{MFR_{CHN^+ + CHON^+ \, T=90°C}} \qquad (1)$$

The ratio of $NO^+$ to $NO_2^+$ of pON (denoted as $R_{ON,calc}$) can be determined as

$$[NO_{3,inorg}] = [NO_3] - [NO_{3,org}] \qquad (2)$$

$$[NO_{inorg}] = \frac{R_{AN}}{1 + R_{AN}} \times [NO_{3,inorg}] \qquad (3)$$

$$[NO_{2,inorg}] = \frac{1}{1 + R_{AN}} \times [NO_{3,inorg}] \qquad (4)$$

$$[NO_{org}] = [NO] - [NO_{inorg}] \qquad (5)$$

$$[NO_{2,org}] = [NO_2] - [NO_{2,inorg}] \qquad (6)$$

$$R_{ON,calc} = \frac{[NO_{org}]}{[NO_{2,org}]} \qquad (7)$$

15   The subscript "$T = 90$ °C" denotes the mass concentrations and mass fraction remaining (MFR) of fragments at $T = 90$ °C. The subscripts "org" and "inorg" denote the mass concentration of fragments of particulate organic and inorganic nitrates. NO, $NO_2$ and $NO_3$ are the measured mass concentrations of total nitrate fragments. $R_{AN}$ is the average ratio of $NO^+$ to $NO_2^+$ from ammonium nitrate.



Note that the pON/OA in this study refers to the contribution of pON to total OA including the contribution of nitrate functional groups (i.e., total OA = Org + $NO_{3,Org}$). The conversion method from $NO_{3,Org}$/Org into pON/OA is illustrated in Takeuchi and Ng (2019).

## 3 Results and discussion

### 3.1 Intercomparisons

Figure 2 shows the $NO_{3,org}$ mass loadings estimated from three different methods during 3 campaigns. The $NO_{3,org}$ loadings calculated by "TD-AMS method" correlated well with those of "$NO_x$ method" during summertime ($R^2$=0.62), but different slopes were found in different periods of the measurements. For example, the slope was 1.1 during 15-17 June ($R^2$=0.94), implying higher $NO_{3,org}$ loadings calculated by "$NO_x$ method" with $R_{ON}$ =15 than that using "TD-AMS method". Comparatively, the slope was 0.35 during 11-12 June ($R^2$=0.62), suggesting lower loadings from the "$NO_x$ method". Such differences were likely due to different pON species that were formed under different precursors and meteorological conditions (e.g., temperature, $NO_x$ levels and RH). Also, the slope differences indicate the variations of pON compounds leading to the variation of $R_{ON}$ ($R_{ON,calc}$=5.0 during 11-12 June vs. 13.6 during 15-17 June), highlighting the importance of determining the time-dependent $R_{ON}$. On average, the mass concentration of $NO_{3,org}$ calculated by "TD-AMS method" in summer was 1.8 µg m$^{-3}$, which was slightly higher than that from "$NO_x$ method" (1.0-1.5 µg m$^{-3}$). Negative $NO_{3,org}$ mass loadings from the "$NO_x$ method" were observed, accounting for 8.5% and 12.8% of the time during wintertime in Beijing and Gucheng, respectively. Such phenomenon was also observed in previous studies in China (Zhu et al., 2016;Yu et al., 2019), particularly in winter and was likely due to (1) high emissions of anthropogenic volatile organic compounds (AVOC) and high $NO_x$ condition resulting in similar $R_{ON}$ and $R_{AN}$, and (2) much higher inorganic nitrate loading than pON. As shown in Fig.2, the $NO_{3,org}$ from the "TD-AMS method" correlated well with that from the "$NO_x$ method", yet the mass loadings were deviated from the lower and upper bounds from the "$NO_x$ method" during some periods likely due to different pON compounds. The $NO_{3,org}$ loadings from the "$NO_x$ method" ranged from 0.9 to 1.3 µg m$^{-3}$ and from 0.8 to 1.6 µg m$^{-3}$ in Beijing and Gucheng during wintertime, respectively, which is overall comparable to that of "TD-AMS method" in Beijing (1.0 µg m$^{-3}$) and slightly lower than that in Gucheng (1.9 µg m$^{-3}$).

The average mass concentration of $NO_{3,org}$ calculated from the "PMF method" (0.28 µg m$^{-3}$) is much lower than those from the other two methods in summer in Beijing, in agreement with the results in Southeastern U.S. (Xu et al., 2015b). In contrast, the average $NO_{3,org}$ (2.5 µg m$^{-3}$) in winter in Beijing was higher, and that (1.8 µg m$^{-3}$) in winter in Gucheng was comparable to that from the other two methods. Such differences during three campaigns indicate the uncertainties of "PMF method" in



estimation of $NO_{3,org}$ can originate from, for example, (1) the incorrect assignment of $NO^+$ and $NO_2^+$ to POA factors, (2) the different ratio of $NO^+$ to $NO_2^+$ in nitrate inorganic aerosol (NIA) factor from pure ammonium nitrate, and (3) the contribution of organic ions to NIA factor. As shown in Fig. S2, the nitrogen-containing fragments ($C_xH_yN^+$, accounting for 5% of NIA) were unexpectedly assigned to NIA factor, while the $NO_x^+$ in MO-OOA, a factor often related to aqueous-phase processing

(Xu et al., 2017) were negligible, resulting in the minor response of $NO_{3,org}$ loadings as a function of RH. This is also supported by the fact that the concentrations of $NO_{3,org}$ from the "PMF method" were much lower than those from "TD-AMS method" under high RH levels (10-12 June, RH = 70%). The $NO_x^+$ signals assigned to POA in winter in Beijing were much higher than those in source emission experiments. For example, $f_{NO+}$ (3.5%) and $f_{NO2+}$ (1.5%, fractions of $NO^+$ or $NO_2^+$ in total mass spectrum) of biomass burning OA in winter in Beijing (Fig.S3) were higher than that from burning of different biofuels ($f_{NO+}$

=0.9% and $f_{NO2+}$=0.65%) (Xu et al., 2020). Further analysis suggested that the episodes dominated by POA also resulted in higher pON loadings from the "PMF method" in winter in Beijing. Considering the uncertainties of "PMF method", the following discussions were mainly based on the results from the "TD-AMS method" and "$NO_x$ method".

The average mass concentration of $NO_{3,org}$ in summer was higher than that in winter in Beijing, and also the contribution of $NO_{3,org}$ to $NO_3$ (24.1% vs. 9.8%). This result is consistent with the seasonal variation of $NO_{3,org}$ loadings observed in the

Southeastern U.S. (Xu et al., 2015b). One explanation is stronger secondary formation during summertime than winter, as indicated by the higher fraction of SOA and SIA (Zhou et al., 2020). The dominant biogenic volatile organic compounds (BVOC) in summer and AVOC in winter (Liu et al., 2020) would be another reason for the differences in pON loadings in two seasons. The contribution of $NO_{3,org}$ to $NO_3$ (9.8% vs. 11.6%) is comparable between Beijing and Gucheng during wintertime, yet the average $NO_{3,org}$ in Gucheng was higher than that in Beijing during wintertime. This can be attributed to much higher

anthropogenic emissions at the rural site than urban site, resulting in different pON compounds as suggested by the different $R_{ON}$ ($R_{ON,calc}$=2.2 in Beijing and 4.3 in Gucheng on average). Also, the variations of N mass contributions from different N-containing ions as a function of TD temperature supported the differences in pON compounds (Fig. 1). Note that the contribution of pON (assuming $MW_{ON}$=200 g mol$^{-1}$) to OA in NCP (14.8%-27.5%) was in the range of previous studies (Ng et al., 2017;Yu et al., 2019), and the seasonal variations of pON/OA in Beijing (14.8% in winter vs. 27.5% in summer) were

also similar to previous studies (Xu et al., 2015b). The higher contribution of pON to OA in Gucheng than Beijing (24.0% vs. 14.8%) indicated more important role of pON in OA at the rural site in NCP during wintertime.

The negligible correlation between $NO_{3,org}$ with POA in summer in Beijing ($R^2$=0.05) is in contradiction with the results in Xianghe (a rural site in NCP), which shows that pON formation might be highly associated with primary emissions (Qiao et al., 2020). Such differences were caused by the different primary emission in different environments. For example, BBOA is

the dominant POA in Xianghe (Qiao et al., 2020), but not in Beijing during summertime. The higher correlation between



$NO_{3,org}$ with POA was found in winter in NCP ($R^2$=0.49-0.52), suggesting that primary emissions can be an important source of pON in winter in NCP, which may be related to $NO_3$ oxidation of AVOC emitted from anthropogenic activities (Li et al., 2019;Lee et al., 2019). Our results are consistent with the considerable contributions of alkanes (a typical AVOC) to pON production in previous field (Lee et al., 2015), model (Jordan et al., 2008) and chamber studies (Lim and Ziemann, 2009). In

fact, there is a growing consensus about the importance of AVOC (e.g. styrene (Yu et al., 2019), toluene (Ramasamy et al., 2019) and long-chain aliphatics (Matsunaga and Ziemann, 2010)) in pON formation under high $NO_x$ condition (Lee et al., 2015;Li et al., 2019;Lee et al., 2019).

## 3.2  Diurnal variation

As shown in Fig.3, the $NO_{3,org}$ loadings and contributions from the "TD-AMS method" and "$NO_x$ method" showed similar

diurnal profiles in summer in Beijing, which were characterized by decreased concentration during daytime, in agreement with the behaviors of total $NO_3$ in Beijing (Sun et al., 2013) and $NO_{3,org}$ in Shenzhen (Yu et al., 2019). The elevated planetary boundary layer (PBL) and temperature-dependent gas–particle partitioning could be the major reason. We noticed an elevated fraction of $NO_{3,org}$ in $NO_3$ during daytime in summer, likely suggesting that photochemical processing played an important role in the formation of pON in summer in Beijing. This is consistent with the moderate correlation of $O_x$ with $NO_{3,org}$ during 12:00-

18:00, and tight correlation between LO-OOA, an OA factor related to photochemical production (Xu et al., 2017) and $NO_{3,org}$ from 6:00-16:00 during summertime (Fig. S5). Note that $NO_{3,org}$ loadings, the contributions of pON to OA, and $NO_{3,org}/NO_3$ showed apparent increases starting at 20:00 in summer, highlighting the important role of $NO_3$ radical in the nocturnal pON formation (Fry et al., 2013;Zhang et al., 2016;Ng et al., 2017). Indeed, we calculated the ratio of $NO^+$ to $NO_2^+$ of ON during daytime and nighttime and found higher $R_{ON}$ at night ($R_{ON,calc}$=5.8) than daytime ($R_{ON,calc}$=5.2) in summer, suggesting that the

pON compounds from OH-initiated and $NO_3$-initiated chemical reactions were different. This higher $R_{ON}$ for pON formed via $NO_3$ oxidation than those via OH oxidation is consistent with the result of chamber experiments (Takeuchi and Ng, 2019).

The diurnal variations of $NO_{3,org}$ loadings in winter in Beijing and Gucheng were characterized by significant increases during daytime, indicating that photochemical production played an important role in pON formation. However, the fraction of pON in OA remained relatively stable throughout the day, suggesting that photochemical processing also produced other SOA

besides pON. $NO_{3,org}$ from the two different methods showed some different trend during some periods in winter (Fig.3). For example, the $NO_{3,org}$ from the "$NO_x$ method" dropped at 8:00, while that from "TD-AMS method" peaked at 8:00 in winter in Beijing. The differences in $NO_{3,org}$ from the two different methods also appeared in the morning in Gucheng, where $NO_{3,org}$ loadings from the "TD-AMS method" showed an obvious enhancement at ~6:00, yet not in "$NO_x$ method". Also, the fraction of pON from the "TD-AMS method" in OA (22.8%) was higher than that from "$NO_x$ method" (5.7-11.0%).Such differences



in the morning during wintertime could be attributed to the differences in organic nitrogen composition with different volatility and also different $R_{ON}$ between morning and afternoon. In Beijing in winter, the peak of $NO_{3,org}$ from the "TD-AMS method" was partly caused by the changes in AVOC emissions (e.g., acetone, benzene, c8-aromatics and toluene) (Sheng et al., 2019). In Gucheng in winter, photochemical aqueous-phase processing was one of the reasons for those differences in the morning

due to the frequent occurrence of radiation fog that dissipated after sunrise (Kuang et al., 2020). Overall, our results suggest that the use of a fixed $R_{ON}$ value in "$NO_x$ method" was limited in representing diurnal variation characteristics of pON in winter in NCP.

### 3.3 Dependence of pON on NR-PM₁ loading

As shown in Fig.4, $NO_{3,org}$ showed an overall increasing trend as the increase of PM in both summer and winter. However, the

variations of pON in OA were different between summer and winter. The pON fraction in OA were comparable (~14%) between summer and winter in Beijing in clean days (NR-PM$_1$ < 20 µg m$^{-3}$), yet the $R_{ON}$ in summer ($R_{ON,calc}$=7.0) was much higher than that in winter ($R_{ON,calc}$=2.8), emphasizing the seasonal differences in pON composition. Comparatively, the pON fraction in OA at the rural site was higher than that in Beijing in winter, and the $R_{ON}$ was also different ($R_{ON,calc}$=6.0 in Gucheng vs. $R_{ON,calc}$=2.8 in Beijing) highlighting the very different pON compounds between rural and urban sites during clean days.

The fraction of pON in OA at high PM levels shows significant differences. For example, the fraction of pON in OA showed a clear increase as a function of PM and reached ~30% at high PM level (> 50 µg m$^{-3}$) in summer in Beijing, indicating the increased contribution of pON to OA during polluted episodes. Comparatively, the contribution of pON to OA remained small under different PM levels in winter in Beijing and Gucheng. By linking the variations of pON with OA factors, we found that LO-OOA (a factor related to the photochemical processing (Xu et al., 2017)) shows obvious increase in polluted conditions in

summer, yet not in winter in Beijing (Fig. S6), indicating that photochemical processing played an increasing role in polluted conditions during summer yet limited in winter in Beijing. In addition, the largely enhanced primary emissions at high PM levels in winter could be another reason for the lower contribution of pON to OA. Different from the variations of pON/OA, the fraction of $NO_{3,org}$ in total $NO_3$ showed overall decreasing trends as a function of PM levels, indicating that the increases of inorganic nitrate were more rapid than $NO_{3,org}$. We found that $NO_{3,org}$ was more important than inorganic nitrate during clean

days in summer by contributing ~80% although the contribution decreased rapidly to 20% in polluted episodes.

Table 1 summarizes the average $R_{ON,calc}$ in clean and polluted days during three campaigns. $R_{ON,calc}$ showed an lower value in polluted days than that in clean days, indicating the changes of ON compounds in different PM levels. Note that the $R_{ON,calc}$ of 1.2 in polluted days in winter in Beijing was lower than the $R_{AN}$ of 2.8. Besides the intense primary emission, complicated precursors in winter, the mixture effect from AVOC and BVOC oxidations may have played a part in this low $R_{ON,calc.}$, which





warrants further studies. In addition, the uncertainties of $R_{AN}$ from a period with high $NO_3$ loadings could be another possible reason.

## 3.4  Effects of meteorological conditions

Figure 5 shows the variations of $NO_{3,org}$ mass loadings as a function of RH and temperature during daytime (8:00-20:00) and nighttime (20:00-8:00) during three campaigns. In general, $NO_{3,org}$ showed an increasing trend as function of RH below 80% in both daytime and nighttime. Previous studies in NCP showed ubiquitously increased PM levels as a function of RH. In addition to the increased aqueous-phase processing, the stagnant meteorological conditions under high RH levels were often found to be the major reason. Therefore, the increases of $NO_{3,org}$ as a function of RH could be mainly associated with the increased PM levels. As RH increased above 90%, we observed some decreases in $NO_{3,org}$, possibly indicating the impact of hydrolysis of pON (Takeuchi and Ng, 2019;Rindelaub et al., 2015;Liu et al., 2012). However, since OA concentration also exhibited a decreasing trend above 80-90% RH as with $NO_{3,org}$, the decrease in $NO_{3,org}$ may have been also affected by partitioning. The day and night differences in $NO_{3,org}$ under low and high RH levels were also shown in Fig.5. For example, we observed higher $NO_{3,org}$ loadings during daytime than nighttime at RH<60%, indicating that photochemical formation of pON was more important than nocturnal production, which is further supported by the elevated fraction of $NO_{3,Org}$ in $NO_3$ in daytime (Fig. 3). However, higher $NO_{3,Org}$ loadings at nighttime than daytime were observed at RH > 60% in Gucheng, likely suggesting the importance of nocturnal reactions associated with enhanced primary emissions at high RH level. Overall, the day-night differences highlight the impacts of multiple factors on ON formation, including precursors, e.g., biogenic and anthropogenic VOCs, and oxidants, e.g., OH and $NO_3$ radicals, which needs to be further investigated by analyzing molecular compositions of ON.

The $NO_{3,org}$ loadings showed a slightly decreasing trend as function of $T$ in summer due to the evaporative loss under high temperature. This result was consistent with the chamber experiments showing decreasing monoterpenes pON as temperature increased from ~25 to ~40 °C (Boyd et al., 2017;Berkemeier et al., 2020). In contrast, the $NO_{3,org}$ loadings in both daytime and nighttime showed increasing trends as a function of $T$ in winter in Beijing suggesting the importance of photochemical production. In contrast, the $NO_{3,org}$ loadings decreased as temperature elevated in Gucheng during wintertime, and then remained relatively stable at $T > 0$°C. One reason was due to much enhanced primary emissions from residential heating under low temperature.

## 4    Limitations and Conclusions

In this study, we demonstrated the capability of combining HR-AMS and TD measurements to quantify and characterize pON

in NCP in summer and winter. However, this approach has several limitations. Firstly, we assumed that $NO_{3,inorg}$ evaporated completely at $T = \sim90\ °C$ , which might not be the case in some environments and could overestimate evaporative loss of $NO^+$ and $NO_2^+$ from ON. Secondly, the average mass loss of $C_xH_yN_z^+$ and $C_xH_yO_zN_p^+$ at $T = 90\ °C$ were used as the proportion of mass loss of pON, however, the volatilities of pON vary among different pON species. In addition, separating nitrogen ion fragments is a challenge due to the limited resolution of HR-AMS, leading to the uncertainties in calculating mass losses of $C_xH_yN_z^+$ and $C_xH_yO_zN_p^+$. Thirdly, using a default value of $RIE_{ON} = 1.4$, and a fixed molar mass ($MW_{ON}=200\ g\ mol^{-1}$) for estimation of pON would introduce additional uncertainties considering the complexity of pON compounds. Fourthly, the low volatility nitrogen-containing compounds (e.g. metal nitrates) with high $NO_x^+$ ratio cannot be separated resulting in an overestimation of pON. Lastly, HR-AMS only measures submicron non-refractory pON, which might also underestimate pON in ambient air.

Overall, the pON estimated from the "TD-AMS method" correlated well with those from PMF and $NO_x$ methods, yet has advantages in estimating pON in anthropogenically dominant environment. Particularly, we found that $R_{ON}$ varied differently between day and night, clean and polluted episodes, highlighting different pON compounds from different chemical pathways and sources. Our estimation showed higher mass concentrations of $NO_{3,org}$ in summer than that in winter (1.8 vs. 1.0 µg m$^{-3}$), and also the contribution of pON to OA (27.5% vs. 14.8%), indicating more pON formation in biogenically and anthropogenically mixed environment. In addition, the $NO_{3,org}$ loadings at the rural site (1.9 µg m$^{-3}$) was higher than that in Beijing due to higher primary emissions and gaseous precursors during wintertime.

***Data availability.*** The data in this study are available from the authors upon request (sunyele@mail.iap.ac.cn).

***Author contributions.*** YS and WX designed the research. WX, CC, YQ, CX, WanX and NM conducted the measurements. WX, MT, CC and YQ analyzed the data. MT, DW and NLN reviewed and commented on the paper. WX and YS wrote the paper.

***Competing interests.*** The authors declare that they have no conflict of interest.

***Acknowledgements.*** This work was supported by the Special research Assistant project of the Chinese Academy of Sciences and China Postdoctoral Science Foundation (2020M670421). MT and NLN acknowledged support from NOAA NA18OAR4310112 and NSF CAREER AGS-1555034.





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





**Table 1. A summary of average $R_{ON,calc}$ during three campaigns**

|  | Beijing$_{Summer}$ | Beijing$_{Winter}$ | Gucheng$_{Winter}$ |
|---|---|---|---|
| NR-PM$_1$ < 20 µg m$^{-3}$ | 7.0 | 2.8 | 6.0 |
| NR-PM$_1$ > 50 µg m$^{-3}$ | 3.4 | 1.2 | 3.5 |
| Average | 5.7 | 2.2 | 4.3 |

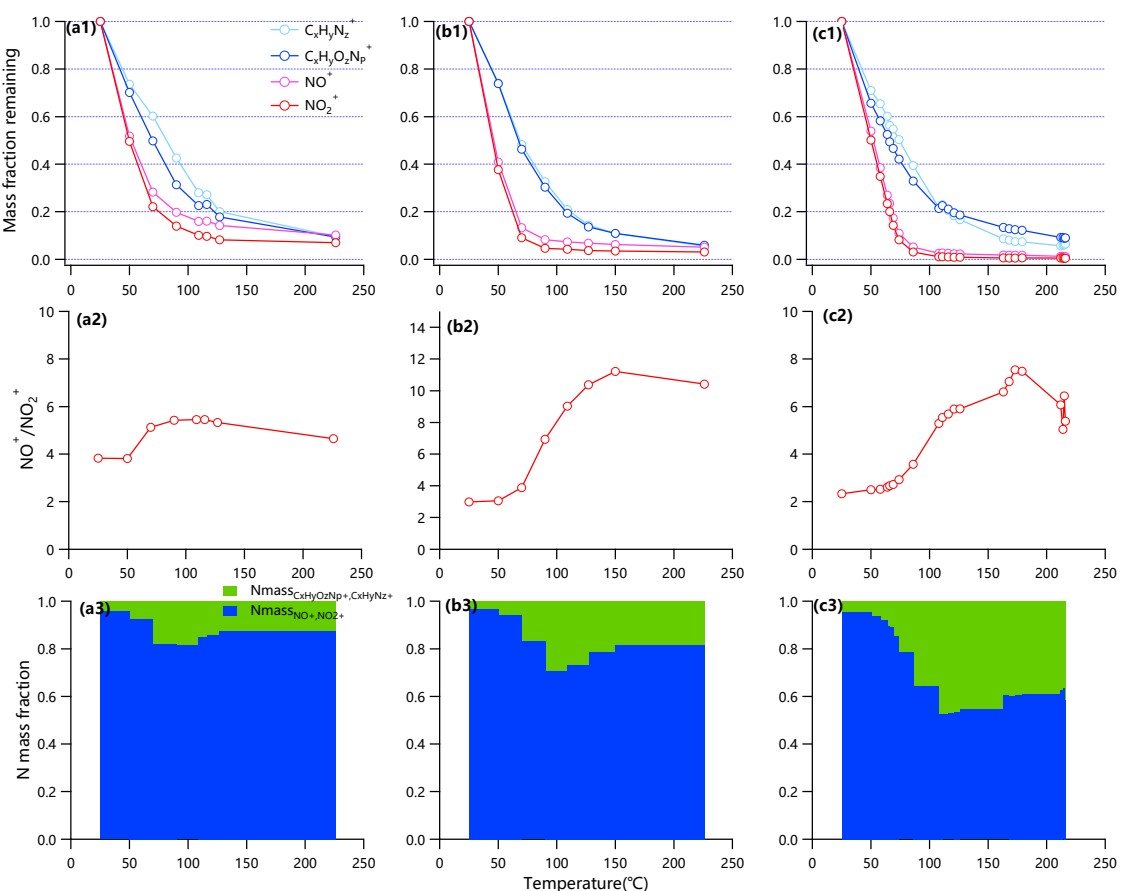

**Figure 1. Thermograms of $C_xH_yN^+$, $C_xH_yON^+$, $NO^+$ and $NO_2^+$ in (a) summer in Beijing, (b) winter in Beijing and (c) winter in Gucheng. Variations of fraction of N mass from N-containing ions, ratio of $NO^+$ and $NO_2^+$ as a function of temperature are also shown.**



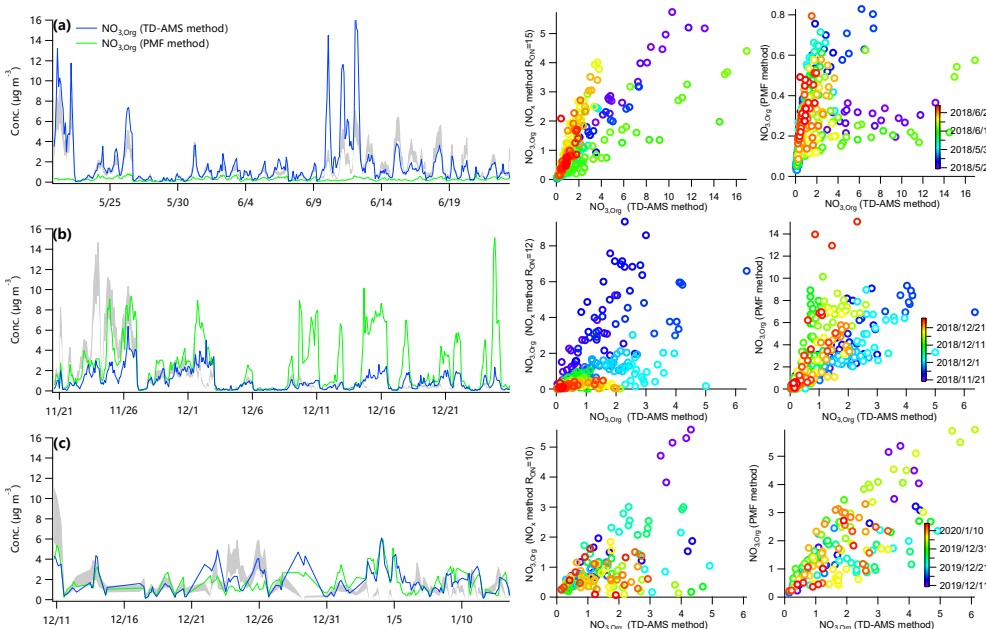

**Figure 2. Intercomparisons of NO$_{3,org}$ mass loadings using three different methods in (a) summer in Beijing, (b) winter in Beijing and (c) Gucheng. The top and bottom of shaded areas in time series are the upper and lower bounds of pON loadings from "NO$_x$ method".**





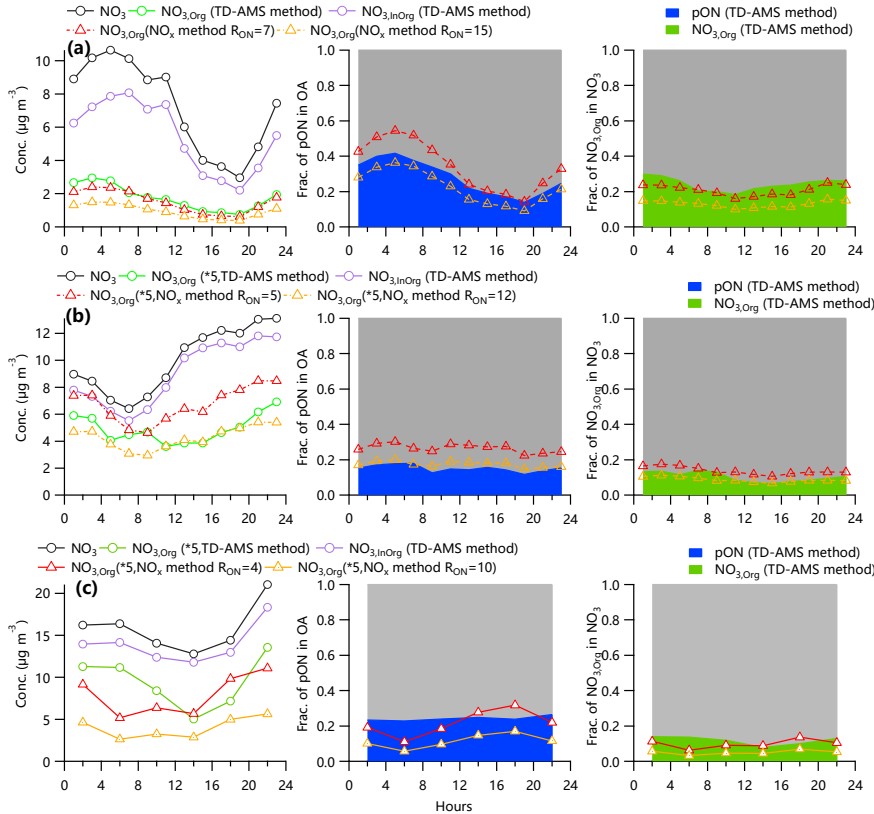

**Figure 3. Diurnal profiles of NO$_{3,org}$ loadings, fraction of pON in OA , and fraction of NO$_{3,org}$ in NO$_3$ calculated by "TD-AMS method" and "NO$_x$ method" in (a) summer in Beijing, (b) winter in    Beijing and (c) Gucheng. Because there are fewer data in Gucheng due to more TD temperature settings, the diurnal profile shows 4-hour averaged data.**



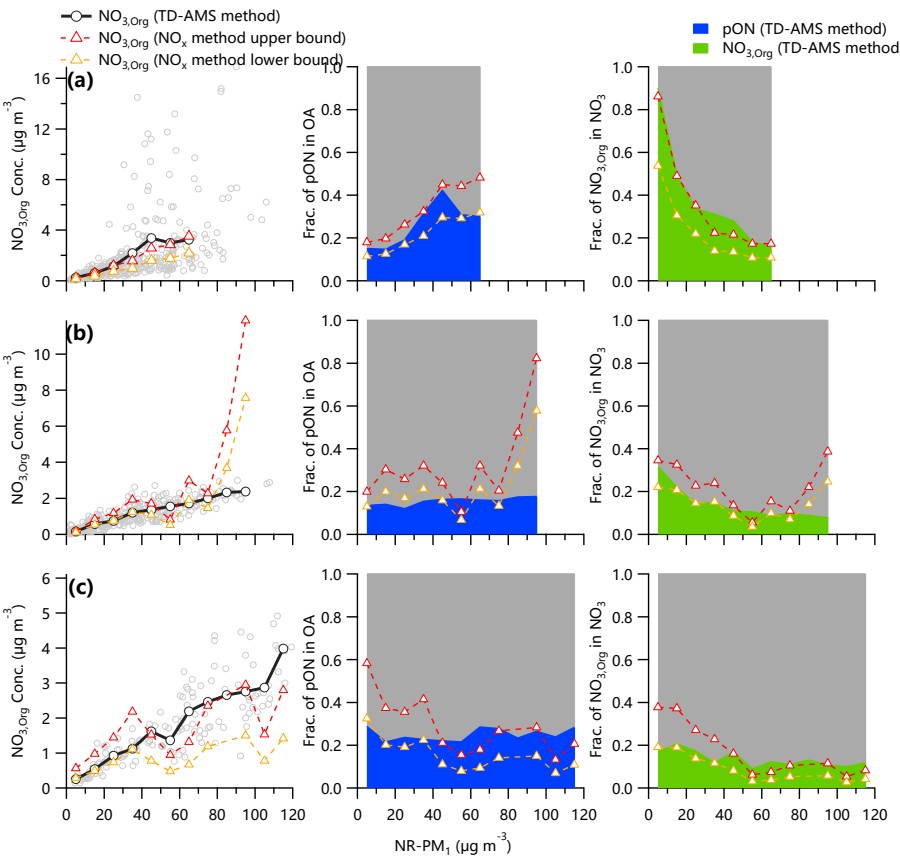

**Figure 4. Variations of NO$_{3,org}$ mass loadings, fraction of pON in OA and fraction of NO$_{3,org}$ in NO$_3$ as a function of particulate matter (PM) loadings in (a) summer in Beijing, and (b) winter in Beijing and (c) Gucheng. Because there are fewer data points in Gucheng due to more TD temperature settings, the chemically resolved PM pollution is 20 μg m$^{-3}$ average.**



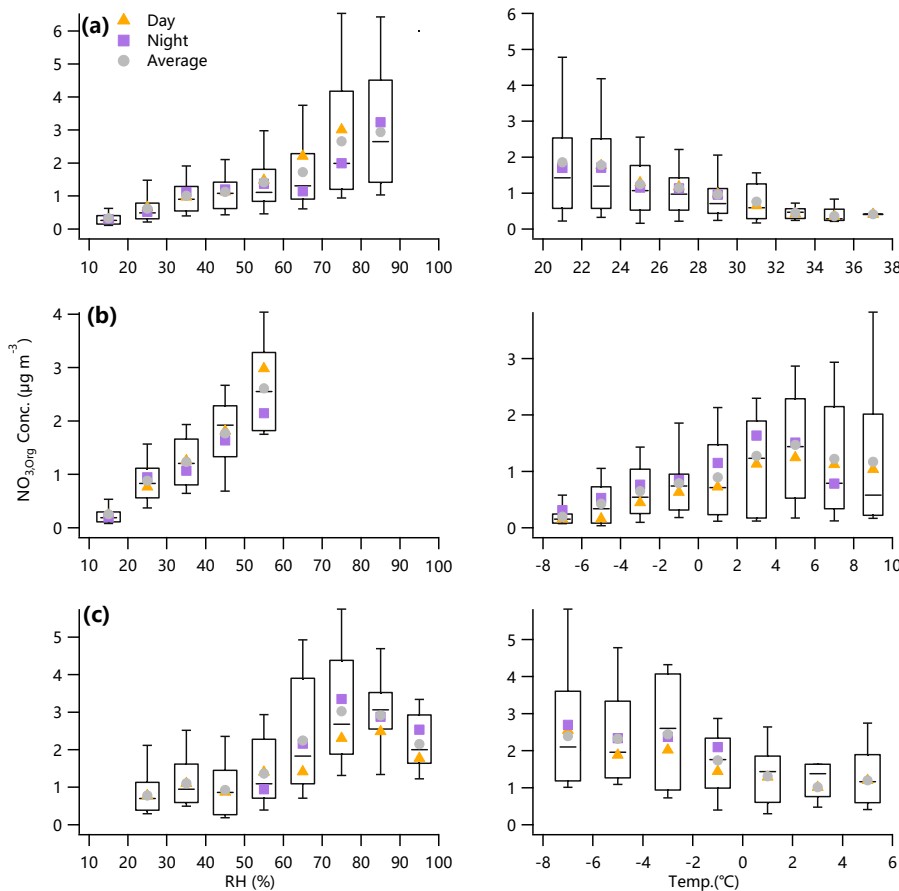

**Figure 5. Variations of NO$_{3,org}$ mass loadings as a function of RH and temperature in (a) summer in Beijing, (b)winter in Beijing and (c) Gucheng calculated by "TD-AMS method". The mid-point line, lower and upper boxes, lower and upper whiskers refer to median, 25th percentiles, 75th percentiles, 10th percentiles, and 90th percentiles, respectively for the entire study.**