# Peer review of "Estimation of particulate organic nitrates from thermodenuderaerosol mass spectrometer measurements in North China Plain"

_Atmospheric Measurement Techniques, 2020_

## Referee Comment (RC1) · Anonymous Referee #1 · 25 Jan 2021

The manuscript by Xu et al. developed a method for estimation of particulate organic nitrates (pON) from the measurements of high-resolution aerosol mass spectrometer coupled with a thermodenuder based on the volatility differences between inorganic nitrate and pON. Generally, the pON loading and pON to OA were compared in detail during three different campaigns in NCP. In addition, $NO_x^+$ ratio of organic nitrates was determined and showed considerable differences between day-night and clean-polluted periods, highlighting the complexity of pON compounds from different chemical pathways (e.g., OH and NO3 oxidation) and sources. The topic fits well within the scope of AMT. This manuscript is generally well written. Before its publication, the following comments need to be addressed.

[Figure]

Specific Comments: 1. I noticed the extremely high NO3,org from the "NOx method" at high RH in Fig.4, yet not in "TD-AMS method". What accounts for this extremely high value? Please elaborate. 2. How about the size distributions of NO3 in different TD temperature? I suppose that if the NO3,inorg evaporated completely at T = 90 °C, the size distributions would be different at T > 90 °C and T < 90 °C. 3. What is the time of a sampling cycle in Gucheng? 4. Are N-containing ions the same in three campaigns? Separation and quantification of N-containing ions are more challenging in V-mode. What are the N-containing ions in CxHyNz+ and CxHyOzNp+? Please elaborate. 5. I suggest adding pON loading and pON to OA in table.1 so that the readers can see them clearly.

---

## Referee Comment (RC2) · Anonymous Referee #2 · 24 Feb 2021

Estimation of particulate organic nitrates from thermodenuder-aerosol mass spectrometer measurements in North China Plain

**General comments:**

This work introduces a combination of thermodenuder technique and high-resolution aerosol mass spectrometry measurements (TD-AMS method) for quantifying particle-phase organic nitrates (pON) in North China Plain. The observations are compared with another two approaches, "NOx method" and "PMF method", that have been used for ambient pON quantification in previous studies. The TD-AMS method does not require any assumption of NO+/NO2+ ratios of pON, which can be significantly different between secondary pON generated from different types of precursors and reaction conditions. The major uncertainties of the TD-AMS method is the assumption of pON volatility. This work fits into the scope of Atmospheric Measurement Techniques although more detail/quantitative discussion on the limitations and uncertainties of the TD-AMS method is required. It will be very beneficial to the scientific community if this work can provide some recommendation/pointers on the selection of appropriate method (among the three methods used in this work) for quantifying pON based on their ambient observations. The current form of discussion is a bit biased toward interpretation of pON formation chemistry and comparison with previous studies. Overall, I recommend this work to be considered for AMT publication after addressing the specific comments below.

**Major comments:**

- Introduction: The potential advantages of TD-AMS method (e.g., temporal variation of RON for better understanding of pON formation chemistry, etc.) that over NOx and PMF methods should be clearly highlighted in the introduction.
- NOx method: Page 4, Lines 2-6: The major reasons for the differences of RAN between pure ammonium nitrate and ambient nitrate should be provided. Does this observation imply that ammonium nitrate was not the major contributor of NOx+ signals during the "high NO3" periods? What were the average organic aerosol mass loadings during the "high NO3" periods and how organic aerosol signals at m/z 30 and 46 may affect the accuracy of NOx+ peak fitting? Please define RAN and RON in this paragraph.
- PMF method: Page 4, second paragraph: (1) although the detailed PMF analysis is not the focus of this work, it is recommended to provide a brief description on how the inclusion of NO+ and NO2+ signals may affect the PMF results interpretation. (2) Lines 13-16: It is unclear whether the reported values of RIE and CE were applied to PMF method only or all the three methods.
- TD-AMS method: (1) The two major assumptions of the TD-AMS method are a) complete evaporation of inorganic nitrates at 90°C and b) the mass fraction remaining of CHN and CHNO fragments equal to that of total pON. To be considered for AMT publication, it is particularly important to provide more quantitative description on the uncertainties of TD-AMS method due to these major assumptions and/or conduct sensitivity tests for the related calculation parameters in order to evaluate the performance of TD-AMS method. (2) Page 4, Lines 25-26: Ambient NOx+ signals can be from both inorganic and organic nitrate so that the argument of pON dominated the total particulate organic nitrogen compounds in NCP is not well supported. (3) Page 5, Lines 3-4: Please elaborate how mixing state of aerosol particles can affect vaporization temperature of inorganic nitrate. (4) Page 5, Line 16-17: The values of RAN were determined by ambient NO+/ NO2+ ratios instead of ammonium nitrate as

discussed in the NOx method. It is unclear whether the values of  $R_{AN}$  were determined in the same way for both NOx and TD-AMS methods. If so, it is misleading to subscript "AN" along the discussion in this section.

- Page 7, Line 1: It seems that the PMF method only include NO+ and NO2+ signals from SOA factors for pON quantification. However, chemical composition of "POA" factors can be affected by atmospheric aging. Please clarify. This also highlight the importance of including some detail of PMF method in the experimental section.
- Page 7, Line 19-10: Please specify the type of anthropogenic emissions at rural site that are much higher than those observed at urban site.
- Page 8, Lines 18-20: Were the averaged RON,Cal values between day and night time significantly different in statistical point of view? Please provide standard deviations for the RON,Cal as well. It is recommended to add diurnal patterns of RON,cal in Figure 3.
- Section 3.3 and Figure 4: Substantial increase of pON was observed at high mass loadings in Beijing winter (Figure 4b). Please elaborate more on this observation.

**Minor comments:**

- Figure S2-S3: The resolution of these figures are too low.
- Page 4, line 25: Please specify the panels of Figure 1 that are referring.

---

## Author Comment (AC1) · 31 Mar 2021

We are thankful to the two referees for their thoughtful and constructive comments which help improve the manuscript substantially. Following the reviewers' suggestions, we have revised the manuscript accordingly. Listed below are our point-by-point responses in blue to each comment that is shown in Italic.

**Response to Reviewer #1**

**General Comments:**

The manuscript by Xu et al. developed a method for estimation of particulate organic nitrates (pON) from the measurements of high-resolution aerosol mass spectrometer coupled with a thermodenuder based on the volatility differences between inorganic nitrate and pON. Generally, the pON loading and pON to OA were compared in detail during three different campaigns in NCP. In addition,  $NO_x^+$  ratio of organic nitrates was determined and showed considerable differences between day-night and clean-polluted periods, highlighting the complexity of pON compounds from different chemical pathways (e.g., OH and NO3 oxidation) and sources. The topic fits well within the scope of AMT. This manuscript is generally well written. Before its publication, the following comments need to be addressed.

We thank the reviewer's comments and have revised the manuscript accordingly.

**Specific Comments:**

1. I noticed the extremely high NO3,org from the "NOx method" at high RH in Fig.4, yet not in "TD-AMS method". What accounts for this extremely high value? Please elaborate.

I assume the "high RH" you mentioned refers to "high PM".

Figure R1. Dependence of RAN vs. Robs on the fraction of NO3,Org in NO3.

Following the methods described in Farmer et al. (2010), the fractional contribution of NO3,Org in NO3 can be determined as:

$$X = \frac{(R_{obs} - R_{AN})(1 + R_{ON})}{(R_{ON} - R_{AN})(1 + R_{obs})}$$
(1)

Eq. 1 can then be rewritten assuming that RON/RAN is a fixed value of a,

$$X = \frac{(R_{obs} - R_{AN})(1 + a \times R_{AN})}{(a \times R_{AN} - R_{AN})(1 + R_{obs})}$$
(2)

We assume that the RAN was in the range of 2 to 4 (upper and lower bounds of RAN due to the fact that AMS was not stabilized), and the Robs varied from 2 to 16, a maximum value in ambient calculated by the RON/RAN = ~4 from  $\beta$ -pinene oxidation experiments multiplying the maximum RAN. In this case, the possible coverage of fraction of NO3,org in NO3 as a function of RON and RAN could be shown in Fig. R1.

Figure R1 shows the dependence of  $R_{AN}$  vs.  $R_{obs}$  on the fraction of  $NO_{3,Org}$  in  $NO_3$  assuming a=3. As shown in Fig. R1, the fraction of  $NO_{3,Org}$  in  $NO_3$  varied from ~0 to ~1 as a function of  $R_{AN}$  at low  $R_{obs}$ , which was larger than that at high  $R_{obs}$ , implying that the fraction of  $NO_{3,Org}$  in  $NO_3$  was sensitive to  $R_{AN}$  at high PM levels with low  $R_{obs}$ .

In this study, RAN was determined from a period with high NO3 loadings assuming the predominant inorganic nitrate contribution. Hence the extremely high ON loadings from the "NOx methods" at high mass loadings were likely due to the uncertainties in pON quantification. The variations of pON compounds due to regional transport and different secondary production at high mass loadings were another possible reason.

Following the reviewer's comments, we added the following description in section 2.2 "Note that large uncertainties in pON quantification were found from the "NOx method" under high PM levels during wintertime due to the assumption for RAN"

2. How about the size distributions of NO3 in different TD temperature? I suppose that if the NO3,inorg evaporated completely at T = 90 °C, the size distributions would be different at T > 90 °C and T

Figure R2. Average size distributions of NO3 at different TD temperatures in (a) summer in Beijing, (b) winter in Beijing and (c) winter in Gucheng.

Figure R2 shows the average size distributions of NO3 at different TD temperatures. The peak diameter of total NO3 showed an obvious decreasing trend as the increase of temperature at  $T < 90^{\circ}$ C in winter in Beijing, which varied from ~450 nm at  $T < 90^{\circ}$ C to ~350 nm at  $T > 90^{\circ}$ C. Such behaviors imply the differences in chemical composition of NO3 between  $T < 90^{\circ}$ C and  $T > 90^{\circ}$ C, consistent with the previous results that organic nitrates were relatively more concentrated at small sizes compared to inorganic nitrates (Yu et al., 2019). Comparatively, the slight differences in variation of peak diameter between different TD temperature were found in summer in Beijing and winter in Gucheng likely due to comparable size distributions between pON and inorganic

nitrates. This was different from the previous results in Shenzhen (Yu et al., 2019), which was partly caused by the differences in pON compounds in different seasons and sampling sites.

*3. What is the time of a sampling cycle in Gucheng?* A complete sampling cycle in Gucheng in winter took 150 min.

Following the reviewer's comments, we added

"while in winter of 2019 in Gucheng, the setting TD temperature ramped linearly from 50 °C to 250 °C with a sampling cycle of 150 min."

4. Are N-containing ions the same in three campaigns? Separation and quantification of N-containing ions are more challenging in V-mode. What are the N-containing ions in  $C_xH_yN_z^+$  and  $C_xH_yO_zN_p^+$ ? Please elaborate.

The N-containing ions are the same in three campaigns. The N-containing ions with exact mass differences from adjacent ions at m/z < 60 (CH4N+, C2H6N+, C3H8N+, CHON+, CH2ON+, CH3ON+ and CH4ON+) were fitted in three campaigns because it is challenging to separate and quantify N-containing ions at m/z > 60 considering that the mass resolution of V-mode is ~2000 (Xu et al., 2017). Previous studies have found that the N-containing ions at large m/z's (>60) accounted for ~20% of the total ON (Xu et al., 2017) which would not affect our conclusions in this study.

Following the reviewer's comments, we added

"Note that the  $C_xH_yN_z^+$  and  $C_xH_yO_zN_p^+$  families refer to the sum of N-containing ions at m/z < 60 due to the limited mass resolution of V-mode (Xu et al., 2017)."

5. I suggest adding pON loading and pON to OA in table, so that the readers can see them clearly

Following the reviewer's comments, we added a summary of average pON and NO3,org loadings, fraction of pON in OA, and fraction of NO3,org in NO3 calculated by "TD-AMS method" during three campaigns in Table S1.

**Response to Reviewer #2**

This work introduces a combination of thermodenuder technique and highresolution aerosol mass spectrometry measurements (TD-AMS method) for quantifying particle-phase organic nitrates (pON) in North China Plain. The observations are compared with another two approaches, "NOx method" and "PMF method", that have been used for ambient pON quantification in previous studies. The TD-AMS method does not require any assumption of NO+/NO2+ ratios of pON, which can be significantly different between secondary pON generated from different types of precursors and reaction conditions. The major uncertainties of the TD-AMS method is the assumption of pON volatility. This work fits into the scope of Atmospheric Measurement Techniques although more detail/quantitative discussion on the limitations and uncertainties of the TD-AMS method is required. It will be very beneficial to the scientific community if this work can provide some recommendation/pointers on the selection of appropriate method (among the three methods used in this work) for quantifying pON based on their ambient observations. The current form of discussion is a bit biased toward interpretation of pON formation chemistry and comparison with previous studies. Overall, I recommend this work to be considered for AMT publication after addressing the specific comments below.

We thank the reviewer's comments and have revised the manuscript accordingly.

**Major Comments**

1, Introduction: The potential advantages of TD-AMS method (e.g., temporal variation of  $R_{ON}$  for better understanding of pON formation chemistry, etc.) that over  $NO_x$  and PMF methods should be clearly highlighted in the introduction. Following the reviewer's comments, we added

"Nevertheless, previous field studies failed to explore the temporal variation of RON due to the limitation of methods and the diversity of pON."

"The temporal variation of RON during three campaigns are elucidated."

2. NOx method: Page 4, Lines 2-6: The major reasons for the differences of  $R_{AN}$  between pure ammonium nitrate and ambient nitrate should be provided. Does this observation imply that ammonium nitrate was not the major contributor of NOx+ signals during the "high NO3" periods? What were the average organic aerosol mass loadings during the "high NO3" periods and how organic aerosol signals at m/z 30 and 46 may affect the accuracy of NOx+ peak fitting? Please define  $R_{AN}$  and  $R_{ON}$  in this paragraph.

Figure R3. Time series of NO3 loadings in ambient and at T=90°C

The cause of the differences in RAN between pure ammonium nitrate and ambient nitrate is not clear yet. One possible reason is that the AMS was not stabilized yet when doing

IE calibration at the beginning of the campaign. For example, we also found the change of  $R_{AN}$  from 4.2 at the beginning of the campaign to 3.4 at the end of the observation in summer of 2017. Unfortunately, we did not do calibrations after the campaigns in Beijing and Gucheng in this study.

Figure R3 shows the time series of the NO3 measured by AMS in ambient and at T=90°C. If ammonium nitrate was not the major contributor of NOx+ signals, the ratio of NO3 at T=90°C to that in ambient would be relatively high due to low evaporative loss of pON. As shown in Figure R2, the ratio of NO3 at T=90°C to that in ambient was low during periods with "high NO3", suggesting the predominant contribution of inorganic nitrate.

---

## Author Comment (AC2) · 31 Mar 2021

The comment was uploaded in the form of a supplement:

Please also note the supplement to this comment:
https://amt.copernicus.org/preprints/amt-2020-474/amt-2020-474-AC2-supplement.pdf

---

## Author Response (AR1)

We are thankful to the two referees for their thoughtful and constructive comments which help improve the manuscript substantially. Following the reviewers' suggestions, we have revised the manuscript accordingly. Listed below are our point-by-point responses in blue to each comment that is shown in Italic.

**Response to Reviewer #1**

*General Comments:*

*The manuscript by Xu et al. developed a method for estimation of particulate organic nitrates (pON) from the measurements of high-resolution aerosol mass spectrometer coupled with a thermodenuder based on the volatility differences between inorganic nitrate and pON. Generally, the pON loading and pON to OA were compared in detail during three different campaigns in NCP. In addition, $NO_x^+$ ratio of organic nitrates was determined and showed considerable differences between day-night and clean-polluted periods, highlighting the complexity of pON compounds from different chemical pathways (e.g., OH and $NO_3$ oxidation) and sources. The topic fits well within the scope of AMT. This manuscript is generally well written. Before its publication, the following comments need to be addressed.*

We thank the reviewer's comments and have revised the manuscript accordingly.

*Specific Comments:*
*1. I noticed the extremely high $NO_{3,org}$ from the "$NO_x$ method" at high RH in Fig.4, yet not in "TD-AMS method". What accounts for this extremely high value? Please elaborate.*

I assume the "high RH" you mentioned refers to "high PM".

[Figure]

Figure R1. Dependence of $R_{AN}$ vs. $R_{obs}$ on the fraction of $NO_{3,Org}$ in $NO_3$.

Following the methods described in Farmer et al. (2010), the fractional contribution of $NO_{3,Org}$ in $NO_3$ can be determined as:

$$X = \frac{(R_{obs} - R_{AN})(1 + R_{ON})}{(R_{ON} - R_{AN})(1 + R_{obs})}$$

(1)

Eq. 1 can then be rewritten assuming that $R_{ON}/R_{AN}$ is a fixed value of a,

$$X = \frac{(R_{obs} - R_{AN})(1 + a \times R_{AN})}{(a \times R_{AN} - R_{AN})(1 + R_{obs})} \quad (2)$$

We assume that the $R_{AN}$ was in the range of 2 to 4 (upper and lower bounds of $R_{AN}$ due to the fact that AMS was not stabilized), and the $R_{obs}$ varied from 2 to 16, a maximum value in ambient calculated by the $R_{ON}/R_{AN} = {\sim}4$ from β-pinene oxidation experiments multiplying the maximum $R_{AN}$. In this case, the possible coverage of fraction of $NO_{3,Org}$ in $NO_3$ as a function of $R_{ON}$ and $R_{AN}$ could be shown in Fig. R1.

Figure R1 shows the dependence of $R_{AN}$ vs. $R_{obs}$ on the fraction of $NO_{3,Org}$ in $NO_3$ assuming a=3. As shown in Fig. R1, the fraction of $NO_{3,Org}$ in $NO_3$ varied from ~0 to ~1 as a function of $R_{AN}$ at low $R_{obs}$, which was larger than that at high $R_{obs}$, implying that the fraction of $NO_{3,Org}$ in $NO_3$ was sensitive to $R_{AN}$ at high PM levels with low $R_{obs}$.

In this study, $R_{AN}$ was determined from a period with high $NO_3$ loadings assuming the predominant inorganic nitrate contribution. Hence the extremely high ON loadings from the "$NO_x$ methods" at high mass loadings were likely due to the uncertainties in pON quantification. The variations of pON compounds due to regional transport and different secondary production at high mass loadings were another possible reason.

Following the reviewer's comments, we added the following description in section 2.2 "Note that large uncertainties in pON quantification were found from the "$NO_x$ method" under high PM levels during wintertime due to the assumption for $R_{AN}$ "

*2. How about the size distributions of $NO_3$ in different TD temperature? I suppose that if the $NO_{3,inorg}$ evaporated completely at T = 90 ℃, the size distributions would be different at T > 90 ℃ and T < 90 ℃*

[Figure]

Figure R2. Average size distributions of $NO_3$ at different TD temperatures in (a) summer in Beijing, (b) winter in Beijing and (c) winter in Gucheng.

Figure R2 shows the average size distributions of $NO_3$ at different TD temperatures. The peak diameter of total $NO_3$ showed an obvious decreasing trend as the increase of temperature at T< 90℃in winter in Beijing, which varied from ~450 nm at T< 90℃ to ~350 nm at T> 90℃. Such behaviors imply the differences in chemical composition of $NO_3$ between T< 90℃ and T> 90℃, consistent with the previous results that organic nitrates were relatively more concentrated at small sizes compared to inorganic nitrates (Yu et al., 2019). Comparatively, the slight differences in variation of peak diameter between different TD temperature were found in summer in Beijing and winter in Gucheng likely due to comparable size distributions between pON and inorganic

nitrates. This was different from the previous results in Shenzhen (Yu et al., 2019), which was partly caused by the differences in pON compounds in different seasons and sampling sites.

*3. What is the time of a sampling cycle in Gucheng?*
A complete sampling cycle in Gucheng in winter took 150 min.

Following the reviewer's comments, we added
"while in winter of 2019 in Gucheng, the setting TD temperature ramped linearly from 50 ℃ to 250 ℃ with a sampling cycle of 150 min."

*4. Are N-containing ions the same in three campaigns? Separation and quantification of N-containing ions are more challenging in V-mode. What are the N-containing ions in $C_xH_yN_z^+$ and $C_xH_yO_zN_p^+$? Please elaborate.*
The N-containing ions are the same in three campaigns. The N-containing ions with exact mass differences from adjacent ions at $m/z < 60$ ($CH_4N^+$, $C_2H_6N^+$, $C_3H_8N^+$, $CHON^+$, $CH_2ON^+$, $CH_3ON^+$ and $CH_4ON^+$) were fitted in three campaigns because it is challenging to separate and quantify N-containing ions at $m/z > 60$ considering that the mass resolution of V-mode is ~2000 (Xu et al., 2017). Previous studies have found that the N-containing ions at large $m/z$'s (>60) accounted for ~20% of the total ON (Xu et al., 2017) which would not affect our conclusions in this study.

Following the reviewer's comments, we added
"Note that the $C_xH_yN_z^+$ and $C_xH_yO_zN_p^+$ families refer to the sum of N-containing ions at $m/z < 60$ due to the limited mass resolution of V-mode (Xu et al., 2017)."

*5. I suggest adding pON loading and pON to OA in table, so that the readers can see them clearly*
Following the reviewer's comments, we added a summary of average pON and $NO_{3,org}$ loadings, fraction of pON in OA, and fraction of $NO_{3,org}$ in $NO_3$ calculated by "TD-AMS method" during three campaigns in Table S1.

**Response to Reviewer #2**
    *This work introduces a combination of thermodenuder technique and high-resolution aerosol mass spectrometry measurements (TD-AMS method) for quantifying particle-phase organic nitrates (pON) in North China Plain. The observations are compared with another two approaches, "$NO_x$ method" and "PMF method", that have been used for ambient pON quantification in previous studies. The TD-AMS method does not require any assumption of $NO^+/NO_2^+$ ratios of pON, which can be significantly different between secondary pON generated from different types of precursors and reaction conditions. The major uncertainties of the TD-AMS method is the assumption of pON volatility. This work fits into the scope of Atmospheric Measurement Techniques*

*although more detail/quantitative discussion on the limitations and uncertainties of the TD-AMS method is required. It will be very beneficial to the scientific community if this work can provide some recommendation/pointers on the selection of appropriate method (among the three methods used in this work) for quantifying pON based on their ambient observations. The current form of discussion is a bit biased toward interpretation of pON formation chemistry and comparison with previous studies. Overall, I recommend this work to be considered for AMT publication after addressing the specific comments below.*

We thank the reviewer's comments and have revised the manuscript accordingly.

*Major Comments*

*1, Introduction: The potential advantages of TD-AMS method (e.g., temporal variation of $R_{ON}$ for better understanding of pON formation chemistry, etc.) that over $NO_x$ and PMF methods should be clearly highlighted in the introduction.*

Following the reviewer's comments, we added

"Nevertheless, previous field studies failed to explore the temporal variation of $R_{ON}$ due to the limitation of methods and the diversity of pON."

"The temporal variation of $R_{ON}$ during three campaigns are elucidated."

*2. $NO_x$ method: Page 4, Lines 2-6: The major reasons for the differences of $R_{AN}$ between pure ammonium nitrate and ambient nitrate should be provided. Does this observation imply that ammonium nitrate was not the major contributor of $NO_x^+$ signals during the "high $NO_3$" periods? What were the average organic aerosol mass loadings during the "high $NO_3$" periods and how organic aerosol signals at m/z 30 and 46 may affect the accuracy of $NO_x^+$ peak fitting? Please define $R_{AN}$ and $R_{ON}$ in this paragraph.*

[Figure]

Figure R3. Time series of $NO_3$ loadings in ambient and at $T$=90°C

The cause of the differences in $R_{AN}$ between pure ammonium nitrate and ambient nitrate is not clear yet. One possible reason is that the AMS was not stabilized yet when doing

IE calibration at the beginning of the campaign. For example, we also found the change of $R_{AN}$ from 4.2 at the beginning of the campaign to 3.4 at the end of the observation in summer of 2017. Unfortunately, we did not do calibrations after the campaigns in Beijing and Gucheng in this study.

Figure R3 shows the time series of the $NO_3$ measured by AMS in ambient and at $T$=90°C. If ammonium nitrate was not the major contributor of $NO_x^+$ signals, the ratio of $NO_3$ at $T$=90°C to that in ambient would be relatively high due to low evaporative loss of pON. As shown in Figure R2, the ratio of $NO_3$ at $T$=90°C to that in ambient was low during periods with "high $NO_3$", suggesting the predominant contribution of inorganic nitrate.

[Figure]

Figure R4. Examples of high resolution mass spectra of *m/z* 30 and 46 in winter in Beijing.

The OA loadings were 23.7 and 36.7 µg m$^{-3}$, respectively during periods with high $NO_3$ in Beijing and Gucheng during wintertime. Figure R3 shows two examples of high-resolution mass spectra of *m/z* 30 and 46 in winter in Beijing. The $CH_2O^+(CH_4N^+)$ and $CH_2O_2^+(CH_4NO^+)$ accounted for 8.2% (<1%) and 8.6% (<1%) at *m/z* 30 and 46, respectively. Therefore, the organic signals at m/z 30 and 46 would not affect the fitting of $NO^+$ and $NO_2^+$ significantly.

$R_{AN}$ and $R_{ON}$ have been defined in introduction as the average ratio of $NO^+$ to $NO_2^+$ from ammonium nitrate and pON, respectively.

*3. PMF method: Page 4, second paragraph: (1) although the detailed PMF analysis is not the focus of this work, it is recommended to provide a brief description on how the inclusion of $NO^+$ and $NO_2^+$ signals may affect the PMF results interpretation. (2) Lines 13-16: It is unclear whether the reported values of RIE and CE were applied to PMF method only or all the three methods.*

We revised it as:
"The mass spectral profiles of all PMF factors resolved from PMF$_{Org+NO3}$ are similar to

those previously resolved in ambient (Yu et al., 2019; Xu et al., 2015b). The mass concentrations of PMF factors resolved from PMF$_{Org+NO3}$ overall were well correlated with that from PMF$_{Org}$ (Table.S2) and also resembled their related tracers. However, some discrepancies in mass concentrations were observed. For example, the concentrations of MO-OOA and LO-OOA resolved from PMF$_{Org}$ were lower than that from PMF$_{Org+NO3}$, which was caused by the signals of NO$_x^+$ in mass spectra of SOA in winter in Beijing."

We also added the Table S2 and listed the contribution of NO$_x^+$ in each factor from PMF$_{Org+NO3}$, ratio and correlation of the concentrations of PMF factors resolved from PMF$_{Org}$ and PMF$_{Org+NO3}$.

The reported values of RIE and CE were applied to all the three methods. We moved the descriptions of RIE and CE to the first paragraph in Section 2.2 to avoid confusion.

*4. TD-AMS method: (1) The two major assumptions of the TD-AMS method are a) complete evaporation of inorganic nitrates at 90 ℃ and b) the mass fraction remaining of CHN and CHNO fragments equal to that of total pON. To be considered for AMT publication, it is particularly important to provide more quantitative description on the uncertainties of TD-AMS method due to these major assumptions and/or conduct sensitivity tests for the related calculation parameters in order to evaluate the performance of TD-AMS method. (2) Page 4, Lines 25-26: Ambient NO$_x^+$ signals can be from both inorganic and organic nitrate so that the argument of pON dominated the total particulate organic nitrogen compounds in NCP is not well supported. (3) Page 5, Lines 3-4: Please elaborate how mixing state of aerosol particles can affect vaporization temperature of inorganic nitrate. (4) Page 5, Line 16-17: The values of R$_{AN}$ were determined by ambient NO$^+$/ NO$_2^+$ ratios instead of ammonium nitrate as discussed in the NO$_x$ method. It is unclear whether the values of R$_{AN}$ were determined in the same way for both NO$_x$ and TD-AMS methods. If so, it is misleading to subscript "AN" along the discussion in this section.*

(1) We agree with the reviewer that it is important to provide more quantitative description on the uncertainties of TD-AMS method. Our first assumption that inorganic nitrate evaporated completely at 90 °C has been well characterized in previous studies. For example, previous studies have found that pure ammonium nitrate evaporated completely at ~50°C (Huffman et al., 2008; Huffman et al., 2009). The uncertainty of our second assumption exists but is challenging to quantify due to the extremely complex pON in ambient air, and also the absence of pON standards. In addition, the organic nitrogen from non-ON compounds also contributed to such uncertainties. Therefore, future studies are needed to further explore such uncertainties. In the revised manuscript, we explained these limitations in more details.

(2) Thanks for pointing this out. We revised it as:
"In addition, the N mass from NO$^+$ and NO$_2^+$ dominated the total N at $T > 90$ ℃, indicating that pON dominated the total particulate organic nitrogen compounds in NCP

in both summer and winter.”

(3) Previous studies have found that secondary inorganic aerosol particles were frequently coated with organics in North China Plain (Li et al., 2021; Liu et al., 2021). Heating such types of particles will evaporate the surface organics first, and then the "core" ammonium nitrate or ammonium sulfate. Therefore, slightly different evaporation temperatures for different ammonium nitrate particles, e.g., internally or externally mixed with organics, are expected.
We revised the sentence as:
For example, Li et al. (2021) and Liu et al. (2021) found that that secondary inorganic aerosol particles were frequently coated with organics in NCP, which may affect the evaporation temperature of AN.

(4) The values of $R_{AN}$ were determined in the same way for both $NO_x$ and TD-AMS methods. $R_{AN}$ refers to the average ratio of $NO^+$ to $NO_2^+$ from ammonium nitrate. However, as we mentioned in Page 4 line 1-6, $R_{AN}$ determined from pure ammonium nitrate in winter was higher than that from ambient observations likely due to the fact that AMS was not stabilized yet when doing IE calibration at the beginning of the campaign. Therefore, we did not use the value in initial calibration but chose a period with high $NO_3$ loadings to determine $R_{AN}$ which were 2.8 and 2.2, respectively in Beijing and Gucheng in winter. Although the $R_{AN}$ of 2.8 and 2.2 were not from pure AN, these values can be considered to represent ammonium nitrate under this assumption. Hence, we called it $R_{AN}$.

*5. Page 7, Line 1: It seems that the PMF method only include $NO^+$ and $NO_2^+$ signals from SOA factors for pON quantification. However, chemical composition of "POA" factors can be affected by atmospheric aging. Please clarify. This also highlight the importance of including some detail of PMF method in the experimental section.*
We thank the reviewer's comments. The pON was calculated as the sum of $NO^+$ and $NO_2^+$ from both POA and SOA factors.
We clarified this and expanded the descriptions of PMF analysis in the revised manuscript.
"The second method is PMF analysis of combined organic and inorganic aerosol (referred to "PMF method") (Sun et al., 2012; Yu et al., 2019; Xu et al., 2015). By including $NO^+$ and $NO_2^+$ ions, PMF analysis was able to separate organic from inorganic nitrates. For instance, the $NO^+$ and $NO_2^+$ ions in the nitrate factor were dominantly from inorganics, and the ratio of $NO^+/NO_2^+$ was close to that of pure ammonium nitrate, while those in OA factors with high $NO^+/NO_2^+$ were generally assumed as organic nitrates (Sun et al., 2012). In this study, the concentrations of $NO_{3,Org}$ are calculated by summing up $NO^+$ and $NO_2^+$ from both POA and SOA factors. "
*6. Page 7, Line 19-10: Please specify the type of anthropogenic emissions at rural site that are much higher than those observed at urban site.*
We revised the sentence as:

"This can be attributed to much higher anthropogenic emissions (e.g., coal combustion and biomass burning) at the rural site than urban site."

*7. Page 8, Lines 18-20: Were the averaged $R_{ON,Cal}$ values between day and night time significantly different in statistical point of view? Please provide standard deviations for the $R_{ON,Cal}$ as well. It is recommended to add diurnal patterns of $R_{ON,Cal}$ in Figure 3.*
Based on Student's t test, the *t* value of 1.79 was larger than that 90% confidence level (1.66) in summer in Beijing, suggesting that the difference between day and night is statistically significant at the 90% confidence level.

[Figure]

Figure R5. Diurnal profiles of $R_{ON,Cal}$ in (a) summer in Beijing, (b) winter in Beijing and (c) Gucheng.

We thank the reviewer's suggestion. Considering that Figure 3 is already crowded, we moved the diurnal profiles of $R_{ON,Cal}$ to supplementary as Fig. S6.

*8. Section 3.3 and Figure 4: Substantial increase of pON was observed at high mass loadings in Beijing winter (Figure 4b). Please elaborate more on this observation.*
We assume that the $R_{AN}$ was in the range of 2 to 4 (upper and lower bounds of $R_{AN}$ due to the fact that AMS was not stabilized), and the $R_{obs}$ varied from 2 to 16, a maximum value in ambient calculated by the $R_{ON}/R_{AN}$ = ~4 from β-pinene oxidation experiments multiplying the maximum $R_{AN}$. In this case, the possible coverage of fraction of $NO_{3,Org}$ in $NO_3$ as a function of $R_{ON}$ and $R_{AN}$ could be shown in Fig. R1.

Figure R1 shows the dependence of $R_{AN}$ vs. $R_{obs}$ on the fraction of $NO_{3,Org}$ in $NO_3$ assuming a=3. As shown in Fig. R1, the fraction of $NO_{3,Org}$ in $NO_3$ varied from ~0 to ~1 as a function of $R_{AN}$ at low $R_{obs}$, which was larger than that at high $R_{obs}$, implying that the fraction of $NO_{3,Org}$ in $NO_3$ was sensitive to $R_{AN}$ at high PM levels with low $R_{obs}$.

In this study, $R_{AN}$ was determined from a period with high $NO_3$ loadings assuming the predominant inorganic nitrate contribution. Hence the extremely high ON loadings from the "$NO_x$ methods" at high mass loadings were likely due to the uncertainties in pON quantification. The variations of pON compounds due to regional transport and different secondary production at high mass loadings were another possible reason.

Following the reviewer's comments, we added the following description in section 2.2 "Note that large uncertainties in pON quantification were found from the "$NO_x$ method""

under high PM levels during wintertime due to the assumption for $R_{AN}$ "

*Minor Comments*
1. *Figures S2-S3: The resolution of these figures are too low.*
Corrected.
2. *Page 4, line 25: Please specify the panels of Figure 1 that are referring.*
Revised.

References

Farmer, D. K., Matsunaga, A., Docherty, K. S., Surratt, J. D., Seinfeld, J. H., Ziemann, P. J., and Jimenez, J. L.: Response of an aerosol mass spectrometer to organonitrates and organosulfates and implications for atmospheric chemistry, Proc. Natl. Acad. Sci. U.S.A., 107, 6670-6675, doi:10.1073/pnas.0912340107, 2010.

Huffman, J. A., Ziemann, P. J., Jayne, J. T., Worsnop, D. R., and Jimenez, J. L.: Development and Characterization of a Fast-Stepping/Scanning Thermodenuder for Chemically-Resolved Aerosol Volatility Measurements, Aerosol Sci. Tech., 42, 395 - 407, 2008.

Huffman, J. A., Docherty, K. S., Aiken, A. C., Cubison, M. J., Ulbrich, I. M., DeCarlo, P. F., Sueper, D., Jayne, J. T., Worsnop, D. R., Ziemann, P. J., and Jimenez, J. L.: Chemically-resolved aerosol volatility measurements from two megacity field studies, Atmos. Chem. Phys., 9, 7161-7182, 2009.

Li, W., Liu, L., Zhang, J., Xu, L., Wang, Y., Sun, Y., and Shi, Z.: Microscopic Evidence for Phase Separation of Organic Species and Inorganic Salts in Fine Ambient Aerosol Particles, Environ. Sci. Technol., 55, 2234-2242, 10.1021/acs.est.0c02333, 2021.

Liu, L., Zhang, J., Zhang, Y., Wang, Y., Xu, L., Yuan, Q., Liu, D., Sun, Y., Fu, P., Shi, Z., and Li, W.: Persistent residential burning-related primary organic particles during wintertime hazes in North China: insights into their aging and optical changes, Atmos. Chem. Phys., 21, 2251-2265, 10.5194/acp-21-2251-2021, 2021.

Sun, Y. L., Zhang, Q., Schwab, J. J., Yang, T., Ng, N. L., and Demerjian, K. L.: Factor analysis of combined organic and inorganic aerosol mass spectra from high resolution aerosol mass spectrometer measurements, Atmos. Chem. Phys., 12, 8537-8551, 10.5194/acp-12-8537-2012, 2012.

Xu, L., Suresh, S., Guo, H., Weber, R. J., and Ng, N. L.: Aerosol characterization over the southeastern United States using high-resolution aerosol mass spectrometry: spatial and seasonal variation of aerosol composition and sources with a focus on organic nitrates, Atmos. Chem. Phys., 15, 7307-7336, 10.5194/acp-15-7307-2015, 2015.

Xu, W., Sun, Y., Wang, Q., Du, W., Zhao, J., Ge, X., Han, T., Zhang, Y., Zhou, W., Li, J., Fu, P., Wang, Z., and Worsnop, D. R.: Seasonal Characterization of Organic Nitrogen in Atmospheric Aerosols Using High Resolution Aerosol Mass Spectrometry in Beijing, China, ACS Earth and Space Chemistry,

10.1021/acsearthspacechem.7b00106, 2017.

Yu, K., Zhu, Q., Du, K., and Huang, X. F.: Characterization of nighttime formation of particulate organic nitrates based on high-resolution aerosol mass spectrometry in an urban atmosphere in China, Atmos. Chem. Phys., 19, 5235-5249, 10.5194/acp-19-5235-2019, 2019.

---

## Author Response (AR3)

We appreciate reviewer #1 for his/her further comments on our manuscript. Following the reviewer's suggestions, we have revised the manuscript accordingly. Listed below are our response to reviewer #1's comments.

**Response to Reviewer #1**

*The authors have addressed most of my major comments in the revised manuscript. Below are a few comments regarding pON quantification by different methods that have to be addressed before publishing in AMT.*

*1) Page 4, NOx method: By assuming most of the NO and NO$_2$ signals observed during high NO$_3$ loadings, R$_{AN}$ of 2.8 and 2.2 are likely higher than that of pure ammonium nitrate due to the presence of pON in ambient. I suggest to provide this information and highlight the potential impacts on pON quantification (e.g. over- or under-determined).*

Following the reviewer's comments, we added the following description in Page 4:
"Note that NO$_{3,org}$ loadings calculated by "NO$_x$ method" were slightly underestimated in winter in this case due to the organic nitrate contribution even in a period with high NO$_3$ loadings."

*2) Page 4, Lines 22: It is somewhat confusing to use "pure ammonium nitrate" here as my understanding is that the R$_{AN}$ values used in this study cannot be obtained by pure ammonium nitrate.*

These descriptions in Page 4, line 20-22 are the previous study in summer in NYC regarding factor analysis of combined organic and inorganic aerosol mass spectra. The R$_{AN}$ values were determined from pure ammonium nitrate.

To avoid confusion, we added the sampling site and time in those descriptions:
"For instance, Sun et al. (2012) performed factor analysis on combined organic and inorganic aerosol mass spectra and found that the NO$^+$ and NO$_2^+$ ions in the nitrate factor were dominantly from inorganics, and the ratio of NO$^+$/NO$_2^+$ was close to that of pure ammonium nitrate, while those in OA factors with high NO$^+$/NO$_2^+$ were generally assumed as organic nitrates in summer in New York City"

*3) Page 5: TD-AMS method: I would like to follow up my previous comment on the TD-AMS method. Can the authors conduct sensitive test (e.g. varying the MFR in equation 1) to evaluate the impacts on pON quantification and RON values from equation 7?*

[Figure]

Figure R1. Dependence of increment of MFR ($R_{AN}$) vs. MFR ($R_{AN}$) on the change percentage of $R_{ON}$ and $NO_{3,Org}$. The summer data were used as initial independent variables in equation 1-7.

Figure R1 shows the possible coverage of the change percentage of $R_{ON}$ and $NO_{3,Org}$ as increment of MFR ($R_{AN}$) and MFR ($R_{AN}$). The increment of MFR was in the range of 0 - 0.5 with a step of 0.01, and the MFR varied from 0 to 1. The $R_{AN}$ varied from ~2 to 5, and the increment of $R_{AN}$ was in the range of 0 - 2.5 with a step of 0.05.

As shown in Fig. R1, the change percentage of $NO_{3,Org}$ and $R_{ON}$ showed overall increasing trends as the rises of increment of MFR at a fixed MFR. The average MFR of the N-containing ions varied from 0.31 to 0.37 during three campaigns, and we assume that the maximal increment of MFR was less than 0.1 (standard deviation of MFR of the N-containing ions at $T$ = 90 °C). In this case, the change percentage of $NO_{3,Org}$ was less than ~20%, suggesting that the impacts of variation of MFR on pON

quantification were relatively low. Comparatively, the change percentage of $R_{ON}$ was relatively high, suggesting the impact of accurate determination of MFR on $R_{ON}$.

We also conducted sensitive test regarding the impact of $R_{AN}$ on $R_{ON}$ values (since the pON quantification was independent of $R_{AN}$). The average $R_{AN}$ varied from 2.2 to 3.8 during three campaigns, and we assume that the maximal increment of $R_{AN}$ was less than 0.5. As shown in Fig. R1, the change percentage of $R_{ON}$ was highly influenced by a $R_{AN}$ rise of less than 0.1, implying the importance of accurate determination of $R_{AN}$ in TD-AMS method.

Following the reviewer's comments, we added:
 "The sensitive tests of $NO_{3,Org}$ and $R_{ON}$ with the variation of MFR and $R_{AN}$ are shown in Fig. S5, demonstrating the impact of accurate determination of MFR and $R_{AN}$ on $R_{ON}$ in "TD-AMS method"."

*4) Page 9, line 16: Refer to my comment #7 in my previous review, it is important to provide the confidence level (90% in this case) in the main text here and to mention diurnal profiles of $R_{ON}$ has been added in the SI.*
Revised.